# Physiological Roles of the Autoantibodies to the 78-Kilodalton Glucose-Regulated Protein (GRP78) in Cancer and Autoimmune Diseases

**DOI:** 10.3390/biomedicines10061222

**Published:** 2022-05-24

**Authors:** Mario Gonzalez-Gronow, Salvatore Vincent Pizzo

**Affiliations:** Department of Pathology, Duke University Medical Center, Durham, NC 27710, USA; salvatore.pizzo@duke.edu

**Keywords:** cancer, autoimmune diseases, ER dysfunctions, GRP78 autoantigenicity, GRP78 cell surface compartments, GRP78 citrullination, GRP78 N-glycosylation

## Abstract

The 78 kDa glucose-regulated protein (GRP78), a member of the 70 kDa heat-shock family of molecular chaperones (HSP70), is essential for the regulation of the unfolded protein response (UPR) resulting from cellular endoplasmic reticulum (ER) stress. During ER stress, GRP78 evades retention mechanisms and is translocated to the cell surface (csGRP78) where it functions as an autoantigen. Autoantibodies to GRP78 appear in prostate, ovarian, gastric, malignant melanoma, and colorectal cancers. They are also found in autoimmune pathologies such as rheumatoid arthritis (RA), neuromyelitis optica (NMO), anti-myelin oligodendrocyte glycoprotein antibody-associated disorder (AMOGAD), Lambert-Eaton myasthenic syndrome (LEMS), multiple sclerosis (MS), neuropsychiatric systemic lupus erythematosus (NPSLE) and type 1 diabetes (T1D). In NMO, MS, and NPSLE these autoantibodies disrupt and move across the blood-brain barrier (BBB), facilitating their entry and that of other pathogenic antibodies to the brain. Although csGRP78 is common in both cancer and autoimmune diseases, there are major differences in the specificity of its autoantibodies. Here, we discuss how ER mechanisms modulate csGRP78 antigenicity and the production of autoantibodies, permitting this chaperone to function as a dual compartmentalized receptor with independent signaling pathways that promote either pro-proliferative or apoptotic signaling, depending on whether the autoantibodies bind csGRP78 N- or C-terminal regions.

## 1. Introduction

GRP78, a member of the 70 kDa heat-shock family of molecular chaperones (HSP70), is a resident of the ER [1]. The ubiquitous localization of GRP78 within the ER is indispensable for the regulation of the UPR resulting from ER stress [2]. During ER stress, GRP78 evades ER retention mechanisms and is translocated to the cell surface, where it functions as a receptor for multiple ligands and also behaves as an autoantigen for autoantibodies that contribute either positively or negatively to several human pathologies such as cancer and autoimmune diseases [3]. Autoantibodies to GRP78 have been identified in patients with prostate, ovarian, gastric, malignant melanoma, and colorectal cancers [4,5]. They are also present in serum from patients with autoimmune pathologies such as RA [6], NMO [7], LEMS [8] and T1D [9]. GRP78 autoantibodies found in the peripheral circulation of MS and NPSLE patients are able to disrupt and move across the BBB reaching the brain [10]. 

The lumen of the ER is an ideal environment for the proper synthesis and folding of proteins destined for secretion or display on the cell surface [11]. Homeostasis in the ER is maintained via coordination of the UPR and ER-associated degradation; however, a variety of disturbances can increase protein misfolding leading to ER stress, where GRP78 initiates signaling cascades that regulate the UPR [12]. Although csGRP78 behaves as an autoantigen in several pathologies, there are major differences in the specificity of its autoantibodies that may be traced down to ER dysfunctions specifically associated with the pathologies of cancer [13] or inflammatory diseases [14]. In this review, we consider some of the mechanisms leading to csGRP78 autoantigenicity and the production of autoantibodies acting as ligands that promote either pro-proliferative or apoptotic signaling depending on whether they bind to either csGRP78 N- or C-terminal domains on the tissues of patients affected by these pathologies. Furthermore, we discuss how ER stress mechanisms modulate the antigenicity of csGRP78 which functions as a compartmentalized receptor with independent signaling pathways.

## 2. Molecular Mechanisms and ER Stress Conditions That Promote GRP78 Autoimmunity in Cancer

GRP78 binds hydrophobic surfaces on newly synthesized polypeptides and is first in line for protein folding, a function that is enhanced when misfolded polypeptides accumulate within the ER as a consequence of cellular stress [2,15]. GRP78 binds to unfolded proteins in its ATP-bound form and mediates their folding at the expense of ATP [16]. When GRP78 functions as a chaperone, it dissociates from the ER transmembrane stress sensor proteins inositol requiring enzyme 1 (Ire1), protein kinase RNA-like ER kinase (PERK) and activating transcription factor 6 (ATF6), which trigger the UPR [17]. Once activated, the UPR signaling promotes the transcription of numerous chaperones to protect the cell from accumulated proteins, and GRP78 itself is a transcriptional target of the UPR via ER stress-responsive elements that can bind to ATF6 [18]. The UPR leads to elevated expression of GRP78 in tumor tissue and controls not only its aberrant localization in the cytosol and the mitochondria but also in the plasma membrane [19]. The over-expression of GRP78 leads to the saturation of the KDEL receptor retrieval mechanism [20], permitting evasion and migration of GRP78 from the ER to these aberrant sites, a phenomenon common to several types of cancer [21]. In prostate cancer and melanoma, csGRP78 induces the production of GRP78 autoantibodies [22,23]. The interaction of these autoantibodies with csGRP78 triggers phosphoinositide 3-kinase (PI3K), protein kinase B (Akt) and mitogen-activated protein kinase (MAPK) signaling pathways, increasing cell survival during tumor growth and metastasis [24,25]. GRP78 is also functionally linked to the immune system via binding to the major histocompatibility complex class I (MHC-I), regulating peptide presentation by MHC-I in the plasma membrane [26]. Overexpression of csGRP78 under ER stress decreases MHC-I expression on the surface [27], and this mechanism is used by tumor cells to evade immune surveillance [28,29]. 

In prostate cancer, the GRP78 autoantibody recognizes a tertiary structure motif with the amino acid sequence CNVKSDKC [30], contained within the GRP78 primary amino acid sequence L^98^IGRTWNDPSVQQDIKFL^115^ (L^98^-L^115^) [22]. This csGRP78 N-terminal region site is also a second receptor for activated α_2_-macroglobulin (α_2_M*) [31] which initiates pro-proliferative signaling pathways upon the binding of either α_2_M* or GRP78 autoantibodies to prostate [22] or melanoma cancer cells [23]. Structural analyses of anti-GRP78 antibodies from a group of patients at different stages of malignant melanoma show that the IgG Fab is asymmetrically glycosylated, while the Fc regions are aberrantly glycosylated [23] in a manner similar to that observed in ovarian cancer [32]. Functionally, the asymmetric glycosylation matters, as reflected by the capacity of these IgG to activate the Akt signaling pathway while promoting the growth of prostate, lung and gastric cancers [33,34]. Asymmetric antibodies have two paratopes, one of high affinity, with K_d_ similar to that of symmetric antibodies and the other one with an affinity for the antigen significantly lower [35]. As a consequence, these IgG function as univalent and fail to trigger some of the major biological reactions of the immune response such as complement fixation, phagocytic activity and antigen clearance, thereby potentially exerting protective immunomodulatory effects on the host tissues [36]. The aberrant glycosylation of the Fc region in the GRP78 Abs in malignant melanoma [23] is the result of a lack of processing of D-mannosyl residues, which leads to a change in the size of the IgG-Fc fragment (27–33 kDa) [37]. These changes decrease their affinity for the Fc receptor (FcR), inducing changes in immune modulation that affect FcR activation [38]. These observations suggest that anti-GRP78 antibodies produced in cancer patients are functionally synthesized to protect the target tissue from immune responses, providing the tumor with defensive mechanisms that facilitate cell survival and proliferation. Figure 1 shows a model summarizing some of the properties and functions that characterize the anti-GRP78 IgG and its associated N-glycan structures identified in cancer.

## 3. Mechanisms and ER Stress Conditions That Promote GRP78 Autoimmunity in RA

RA is a systemic autoimmune disease characterized by chronic inflammation of the joints resulting in hypertrophy of the synovium called pannus, which ultimately leads to joint destruction [39]. The ER response has been implicated in chronic autoimmune inflammatory diseases [12] and ER stressors such as hypoxia, low glucose, and the pro-inflammatory cytokine milieu are responsible for the abnormal proliferation of RA fibroblast-like synoviocytes (RA-FLS) and angiogenesis, promoting pannus formation [40]. During these processes, down-regulation of GRP78 increases apoptosis of RA-FLS, whereas its overexpression prevents R-FLS from apoptotic death induced by ER-stressors [41]. The mechanisms operating in the UPR response of RA cells are similar to those described above for cancer cells [14]; however, during the progress of RA, B cells differentiate into plasma cells and infiltrate the synovial membrane where they synthesize autoantibodies such as immunoglobulin M rheumatoid factor (RF) and anti-cyclic citrullinated peptide antibodies (ACPA) [42,43]. 

Citrullination is a posttranslational modification of proteins catalyzed by peptidylarginine deiminase (PADI) in the presence of a high concentration of calcium that causes a loss of basic charge(s), which can influence the protein structure and create new epitopes recognized by the immune system [44]. In RA patients, the PADI type 4 (PADI4) gene in hematological and synovial tissue is a susceptibility locus for RA that affects the stability of transcripts and is associated with levels of antibody to citrullinated peptides in their sera. The PADI4 haplotype associated with susceptibility to RA increases the production of citrullinated peptides acting as autoantigens in some individuals, leading to a heightened risk of developing the disease, suggesting that genetic factors could cause RA [45]. As a target of PADI4, the citrullinated GRP78 (citGRP78) is arthritogenic in RA mice models, leading to the generation of ACPA which increases the severity of the disease [46]. 

Although the structure of the anti-citGRP78 IgG was not specifically studied in RA, ACPA-IgG molecules show a 10–20 kDa higher molecular weight compared with non-autoreactive IgG, resulting from Fab-linked N-glycosylation that decreases the binding avidity of ACPA for citrullinated antigens [47]. The lower target affinity induced by N-glycosylation of the RA ACPA-IgGs suggests a lower pathogenicity for these antibodies; however, the Fc-linked aberrant decrease in galactosylation and increase in fucosylation patterns observed in RA patients further increases a more pro-inflammatory phenotype of these antibodies [48,49]. The integrity of the IgG Fc region oligosaccharide chain is necessary for the activation of the complement, interactions with the FcR, and antibody-dependent cell-mediated cytotoxicity, all of which are perturbed by truncation of the IgG Asn^297^ oligosaccharide, where hypogalactosylation is a significant factor in the pathogenesis of RA [50].

The epitope specificity of the anti-citGRP78 antibodies in RA was identified using citrullinated peptides in which the arginine residues were replaced with citrulline residues. The RA patient serum antibody levels to GRP78_279–298_ R289citrulline (P12), GRP78_295–314_ R305 citrulline (P15) and GRP78_500–519_ R510citrulline (P23) were significantly increased compared with those of native peptides in RA patients, demonstrating their significance for antibody recognition in RA, in particular the epitope contained in peptide P23 in the GRP78 C-terminal region [41]. The high immunogenicity of the GRP78 C-terminal region was further confirmed in a separate study using a cloned GRP78_409–653_ C-terminal segment, structurally similar to that in the whole protein [51]. The anti-citGRP78 antibodies bind to csGRP78 on monocytes and activate extracellular signal-regulated protein kinases (ERK) and Jun N-terminal Kinase (JNK) pathways, leading to activation of nuclear factor kappa b (NF-κB) and production of tumor necrosis factor alpha (TNF-α) [52,53]. The production of TNF-α further increases csGRP78 expression in RA-FLS, leading to cell apoptosis [14].

## 4. GRP78 Autoantibodies in Immune-Mediated Neurological Diseases

GRP78 autoantibodies have been identified in MS [54], NPSLE [55], AMOGAD [56], LEMS [8], and NMO [57]. MS is a chronic inflammatory demyelinating disease of the central nervous system (CNS), characterized by the presence of focal demyelinated plaques within the white matter [58]. Although the structure of the anti-GRP78 IgG has not been specifically analyzed, the Fc region in IgG1 from MS patients shows reduced fucosylation and galactosylation, and increased bisecting GlcNAc, that promotes a higher pro-inflammatory activity of this immunoglobulin [59].

NPSLE is a complication of SLE characterized by serious cognitive defects which fluctuate over time such as difficulties in attention, concentration and memory [60]. Patients with NPSLE show significantly elevated serum levels of anti-GRP78 antibodies compared with healthy controls [55]. Although the structure of the anti-GRP78 IgG has not been specifically analyzed, significant differences in the IgG glycosylation between SLE patients and controls have been reported [61]. These changes include decreases in the galactosylation and sialylation, which regulate proinflammatory and anti-inflammatory actions of IgG, as well as decreased core fucose and increased bisecting N-acetylglucosamine, which affect antibody-dependent cell-mediated cytotoxicity [61], thereby suggesting that aberrant IgG glycosylation of the anti-GRP78 antibody may be an important pathological mechanism in SLE. 

AMOGAD was recently recognized as a new pathology in the spectrum of inflammatory diseases, which differs from either MS or NMO. The serum levels of anti-GRP78 antibodies in patients with AMOGAD are significantly higher than those of MS patients or healthy controls [56]. LEMS is an autoimmune disease of the neuromuscular junction commonly defined as a paraneoplastic neurological syndrome, involving muscle weakness, areflexia and autonomic dysfunction [62]. Serum levels of anti-GRP78 antibodies of LEMS patients are high, and none were detected in the control subjects [8]. 

NMO is a severe inflammatory autoimmune disorder of the CNS that affects both adults and children. NMO was historically considered a variant of MS [63]; however, the discovery and identification of an NMO IgG autoantibody, specific for aquaporin-4 water channel (AQP4-Ab) [64], facilitated clinical diagnosis and early treatment of NMO [65]. Patient tissue immunopathology demonstrates that astrocytes are the principal cell target in NMO [66]. The circulating AQP4-Abs move across the blood-brain barrier (BBB), reaching brain astrocytes after leakage induced by anti-GRP78 antibodies also present in the circulation of NMO patients [67]. The mechanism by which the anti-GRP78 antibodies break the permeability of the BBB involves their binding to the scGRP78 on the brain microvascular endothelial cells (BMECs), inducing nuclear translocation of NF-κB that facilitates ICAM-1 transcription and promotes binding of activated immune cells that enhance the diameter of cerebral blood vessels in [58]. Similar mechanisms have been observed with anti-GRP78 antibodies in AMOGAD [56] and LEMS [8] that induce brain endothelial cells to open the BBB, thereby allowing access to other pathogenic antibodies in the CNS. A compromised BBB and antiendothelial cell antibodies have been found in patients with SLE [68], suggesting that csGRP78 in brain endothelial cells may also be a target of antiGRP78 antibodies in SLE [69]. The mechanism of disruption of the BBB by GRP78 autoantibodies in NMO, AMOGAD and SLE is summarized in Figure 2.

## 5. GRP78 Autoimmunity in Type 1 Diabetes

T1D is an autoimmune disease characterized by T-cell-mediated destruction of pancreatic β-cells leading to loss of insulin production, unsuppressed glucose production, and hyperglycemia [10]. The β- cells are not a passive target of autoimmunity but actively contribute to their own destruction by triggering the immune system [70] as a result of chronic stress that leads to a switch from a prosurvival to a proapoptotic UPR that causes cell death [71]. Chronic stress stimulates post-translational modifications of β-cell proteins, potentially generating neoepitopes against which no tolerance exists in the immune system [72], such as the citrullination of GRP78 [9], that induces the production of citGRP78 antibodies in patients with T1D [73].

The epitope specificity of the anti-citGRP78 antibodies in patients with T1D is significantly higher against a region containing the peptide I^503^ DVNGIL**R^510^**VTAE^514^ [73]. The same epitope is recognized by citGRP78 antibodies in RA as peptide P23 [41], confirming that the N-terminal region of GRP78 is a target of PAD enzymes, leading to citrullination and autoantigenicity of the same epitopes both in RA [44] and T1D [74], resulting from dysfunctions of the ER common to both autoimmune diseases. Although the specific structure of the anti-citGRP78 IgG glycosylated chains has not been identified in T1D, the analyses show increased agalactosylated IgG glycoforms, suggesting a more proinflammatory IgG profile [75]. 

## 6. Molecular Mechanisms Involved in GRP78 Autoantigenicity and Migration to the Cell Surface

The confirmation that the immune responses of csGRP78 were channeled via two compartments was demonstrated first by the ability of specific N- or C-terminal anti-GRP78 sheep antibodies to induce different calcium signaling waves on csGRP78 prostate cancer cells [4]. Moreover, sections of a human prostatic intraepithelial neoplasm (PIN) high grade were stained with the same anti-GRP78 sheep antibodies against the N- or C-terminal regions and revealed diffuse staining, confirming the availability of both GRP78 regions in prostate cancer tissue [4]. The presence of separate signaling compartments was further demonstrated after incubation of csGRP78 expressing cancer cells with the *E. coli* enzyme subtilase cytotoxin A subunit (SubA), a serine proteinase that cleaves csGRP78 between amino acids Leu^416^ and Leu^417^, releasing a 28-kDa GRP78 C-terminal fragment, that abrogates C-terminal domain signal transduction pathways but keeps the N-terminal-mediated csGRP78 signaling pathways of the cells unaffected [76]. Furthermore, studies in mice with collagen-induced arthritis show that the selective abrogation of GRP78 by SubA, originates a severe impairment of NF-κB and Akt signaling pathways [77].

CsGRP78 is adapted for critical functions related to ER stress in the normal cell; however, alterations in the pathway of trafficking towards the cell surface from the ER, such as those observed in tumoral or autoimmune pathologies, may induce changes in its structural topography, converting GRP78 into a receptor with additional functions not observed in the original ER-linked chaperone, as a result of the localization of its hydrophilic N- and C-terminal regions in the extracellular space {20].

During ER stress in cancer cells, there is an increase in the interaction between the substrate-binding domain of GRP78 and the Gα-interacting vesicle-associated protein, which accelerates cell surface translocation of GRP78 [78]. In addition to the Gα protein, GRP78 is also translocated to the cell surface as a ternary complex between GRP78, the co-chaperone MTJ-1 and Gαq11 associated with lipid rafts/caveolae [4]. The interaction between GRP78 and its partners, MTJ-1 and Gαq11, requires the GRP78 substrate binding site (SBS) activity, whereas ATP binding is not necessary, suggesting that these partner proteins not only accelerate GRP78 translocation to the cell surface but also maintain it on the cell surface [79]. A small population of csGRP78 is tethered by GPI-anchored proteins, making unlikely its transmembrane configuration [79]. These associations separate csGRP78 in two domains that expose the epitope L^98^-L^115^ in the csGRP78 N-terminal region, making it autoantigenic in several types of cancer [4]. The high immunogenicity of this region was confirmed with antibodies raised against soluble recombinant GRP78 in rabbits and sheep which recognize primarily the GRP78 epitope L^98^-L^115^ [4]. The GRP78 autoantibodies bind to this region and promote several signaling pathways involved in cancer cell proliferation and survival [4].

In RA, GRP78 is critical for synoviocyte proliferation and angiogenesis and may be responsible for the transformation of normal synoviocytes into cells with a more aggressive phenotype in RA joints [41]. Rheumatoid B cells recognize GRP78 as an autoantigen, demonstrating that GRP78 expression is higher in infiltrating plasma cells of the RA synovium [80], leading to the synthesis of pathogenic anti-cyclic citrullinated peptide antibodies (ACPA) that recognize citrullinated GRP78 [52]. In RA-FLS, ER stress stimulation of both GRP78 expression and its citrullination leads to a further increase in its autoantigenicity [41]. 

The presence of cit-GRP78 Abs is well characterized in RA and T1D and the epitope specificity of these Abs appears similar in both pathologies. In RA, the region is localized between amino acid residues T^500^FEIDVNGIL**R^510^**VTAEDKGTG^519^, and in T1D the region is localized between amino acid residues I^503^DVNGIL**R^510^**VTAE^514^, where the replacement of **R^510^** by an L-citrulline residue, is preferentially recognized by anti-citGRP78 antibodies [52]. The region containing **R^510^** in the GRP78 C-terminal domain is localized in the hydrophilic side of the ER microsomal transmembrane or in the extracellular space at the cell surface [20] which is accessible to post-translational modifications such as citrullination, common to both pathologies [44,73]. Protein citrullination as a source of neoantigens was recently evaluated in 196 cancer cell lines, but the identification of specific responses against GRP78 was not investigated [81].

## 7. Antibodies and Synthetic Peptides Targeting GRP78 for Therapeutic Purposes

To date, several strategies have been explored to design therapeutic antibodies targeting csGRP78. Several commercial polyclonal antibodies recognizing the C-terminus of GRP78 induce apoptosis or decrease the proliferation of cancer effects [82]. In vitro studies with a recombinant GRP78 antibody (AEP8587) show that this antibody competes with a prostate cancer patient-derived GRP78 autoantibody, decreasing the induction of UPR and tissue factor-procoagulation activity associated with prostate cancer progression [83]. Several commercial polyclonal antibodies recognizing the C-terminal part of GRP78 induce apoptosis or decrease the proliferation of cancer cells [84]. The only GRP78 antibody evaluated in clinical trials is the human IgM antibody named SAM-6, originally isolated from a gastric cancer patient [85]. This antibody showed a strong anti-tumoral effect when initially tested in melanoma-bearing mice, but when evaluated in a phase I clinical trial in a group of patients with malignant melanoma, the results did not show a significant clinical effect [85]. After this trial, there are no reports in the literature on the use of SAM-6 or any other antibody targeting csGRP78. CsGRP78 is also a high-affinity receptor for isthmin (ISM), a protein composed of a central thrombospondin type I repeat (TSR) and a C-terminal adhesion associated domain (AMOP) responsible for the induction of apoptosis in highly metastatic and aggressive cancer cells [86]. Kao et al. used the AMOP domain to design cyclic peptides to target GRP78 [87]. One of such peptides, BC71, suppressed xenograft tumor growth in mice as a single agent after binding to the N-terminal portion of GRP78, an effect antagonized by N-terminal GRP78 antibodies [87]. BC71 can be used not only as a prototype molecule for further development of GRP78 targeted peptidomimetic anticancer therapeutics but also as a radiolabeled molecule in positron emission tomography (PET) imaging to determine csGRP78 levels for cancer prognosis.

## 8. Conclusions

During cellular ER stress, GRP78 evades ER retention mechanisms and is translocated to the cell surface, where it behaves as an autoantigen. GRP78 autoantibodies are found in patients with prostate, ovarian, gastric, malignant melanoma, and colorectal cancers. They are also found in patients with inflammatory autoimmune pathologies such as RA, MS, NPSLE, AMOGAD, NMO, LEMS and T1D. 

In cancer, the anti-GRP78 antibodies recognize the linear GRP78 primary amino acid sequence L^98^-L^115^. In melanoma or ovarian cancer patients the anti-GRP78 antibodies show an IgG Fab asymmetrically glycosylated, while the Fc regions are aberrantly glycosylated. The Fab asymmetric glycosylation decreases the affinity of the Ab for its antigen and impairs its biological functions related to complement fixation, phagocytic activity and antigen clearance, while the Fc aberrant glycosylation reduces the affinity of the anti-GRP78 IgG for the FcR, decreasing its capacity for immune modulation and FcR activation. The changes in N-glycosylation of the anti-GRP78 IgG suggest that they are functionally synthesized to provide cancer cells with mechanisms that facilitate cell survival and proliferation.

In RA, genetic susceptibility increases citrullination of proteins catalyzed by PADI, as a result of the presence of a high concentration of calcium induced by ER stress, promoting the production of anti-citGRP78 antibodies. These antibodies have a Fab-linked N-glycosylation that impairs its binding to citrullinated antigens, suggesting lower pathogenicity. Nevertheless, they display an Fc-linked hypogalactosylation and increase in fucosylation that promotes truncation of the Fc-linked oligosaccharide that exacerbates the pathology of RA. The anti-citGRP78 antibody recognizes an epitope in the C-terminal region containing the amino acid sequence T^500^-Cit^510^-G^519^. The binding of the antibody to this region activates extracellular ERK and JNK pathways, leading to the activation of NF-κB and production of TNF-α which further increases csGRP78 expression in RA-FLS, which unlike the mechanisms observed in cancer, leads the RA cells to apoptosis.

The N-glycosylation structure of the MS anti-GRP78 IgG has not been specifically analyzed, but the Fc region of IgG1 shows reduced fucosylation and galactosylation, and increased bisecting GlcNAc, which promotes a higher pro-inflammatory activity. The SLE IgGs show decreases in the galactosylation and sialylation, which regulate their proinflammatory and anti-inflammatory, while a decreased core fucose and increased bisecting N-acetylglucosamine modify antibody-dependent cell-mediated cytotoxicity. The structure or functions of serum anti-GRP78 antibodies in AMOGAD or LEMS patients are unknown. The NMO anti-GRP78 antibodies break the permeability of the BBB after binding to csGRP78 on BMECs, increasing the diameter of cerebral blood vessels that facilitate the passage of pathogenic autoantibodies from the peripheral circulation to the brain. Similar mechanisms have been observed in AMOGAD, LEMS and SLE.

ER stress in T1D stimulates post-translational modifications of β-cell proteins such as citrullination of GRP78, inducing the production of citGRP78 antibodies. The affinity of T1D anti-citGRP78 antibodies for citGRP78 is significantly higher for a region containing the sequence I^503^-**Cit^510^** -E^514^. The same epitope is recognized by citGRP78 antibodies in RA, confirming that the N-terminal region of GRP78 is a target of PADI enzymes, leading to citrullination and autoantigenicity of an epitope common to both RA and T1D. The structure of the anti-citGRP78 IgG N-glycosylated chains has not been specifically studied in T1D; however, the presence of agalactosylated IgG glycoforms suggests a proinflammatory profile. Therefore, the physiological activities of the anti-GRP78 antibodies are determined not only by their epitope specificity but also structurally by their core N-glycosylation that either attenuates or enhances its proliferative or apoptotic activities, as observed in cancer and immune diseases.

Although csGRP78 has been targeted with exogenous antibodies or peptides with therapeutic potential, the clinical trials did not show a significant clinical effect. Therefore, these agents are still part of a developing field that will require extensive testing before they are considered for clinical treatment of the pathologies discussed in this report.

## Figures and Tables

**Figure 1 biomedicines-10-01222-f001:**
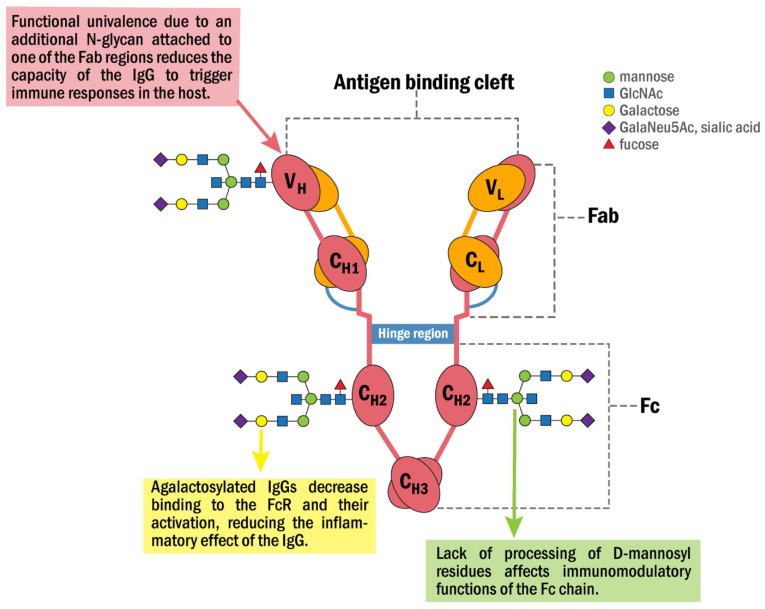
Schematic representation of the anti-GRP78 IgG molecular structure and its associated N-glycan structures identified in cancer patients. It consists of two interconnected heavy (H) and two light chains (L) with two domains that infer different properties, the fragment antigen-binding (Fab) and fragment crystallizable (Fc) domains. The Fab domain is responsible for recognizing and binding the antigen. The Fc domain contains two glycans attached to Asn^297^ in conserved regions of the C_H_2 domain. The Fc domain binds to the FcR on natural killers and other inflammatory cells. In cancer, the Fab region is asymmetrically glycosylated with one additional N-glycan that converts the anti-GRP78 IgG into a univalent molecule that binds the antigen with a decreased affinity, preventing it from triggering some IgG-linked functions of the immune response such as complement fixation, phagocytic activity and antigen clearance. Moreover, the N-glycans of the Fc region are aberrantly glycosylated, mainly at the levels of D-mannosylation and galactosylation with a reduction in the affinity of the antibody for the FcR and its activation, thus inhibiting the inflammatory effects of the IgG.

**Figure 2 biomedicines-10-01222-f002:**
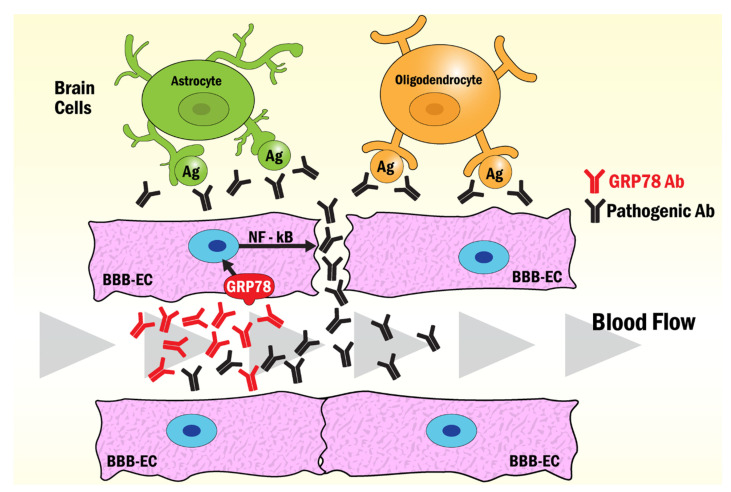
Model of the mechanism of BBB opening induced by GRP78 antibodies common to NMO, AMOGAD, LEMS and SLE. The binding of antibodies to GRP78 on the surface of BBB-endothelial cells induces the activation of NF-κB signaling pathways that disrupt the tight junction enhancing the diffusion of pathogenic antibodies into the brain. Astrocytes react with AQP4-Abs in NMO, while oligodendrocytes react with pathogenic anti-myelin Abs in AMOGAD. Pathogenic antibodies to brain antigens found in LEMS and SLE may also function through this mechanism to induce the damage observed in these pathologies.

## Data Availability

Not applicable.

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
