# Peer review of "Physiological Roles of the Autoantibodies to the 78-Kilodalton Glucose-Regulated Protein (GRP78) in Cancer and Autoimmune Diseases"

_biomedicines, 2022, doi:10.3390/biomedicines10061222_

Round 1
Reviewer 1 Report
Title: Physiological roles of the autoantibodies to the 78-Kilodalton glucose-regulated protein (GRP78) in cancer and autoimmune diseases
Authors: Mario Gonzalez-Gronow, Salvatore Vincent Pizzo
REVIEWER'S COMMENTS:
The Authors have submitted an interesting review paper dedicated to the important topic, namely the roles of anti-csGRP78 autoantibodies in cancer and autoimmune disorders. This material will be helpful for readers of Biomedicines, particularly, for immunologists and oncologists. For my part, there is only a minor criticism. I would recommend to clarify or discuss such points:
What about the specificity/cross-reactivity of the described anti-csGRP78 autoantibodies? (I mean whether these autoantibodies cross-react with some other autoantigens different from GRP78?);
What happens with these autoantibodies after their binding to csGRP78? (Whether the formed immune complexes are internalized into the target cell? If yes, what next?)
CsGRP78 is thought to play an important role in the pathogenesis-promoting activities of cancer cells and particularly, cancer stem cells. How the described anti-csGRP78 autoantibodies affect the cancer-promoting function of csGRP78 in cancer cells and cancer stem cells?
Use of exogenous csGRP78-neutralizing antibodies is discussed as a potential way for treating cancer (see doi: 10.3390/cells9040892; doi: 10.1016/j.canlet.2021.10.004). Whether the presence of anti-csGRP78 autoantibodies will interfere with the use of exogenous (curative) antibodies to csGRP78?
If the Authors consider the anti-csGRP78 autoantibodies as the pathogenic factors, they should discuss some potential ways to overcome such factors (e.g. exogenous antibodies or peptides binding csGR78 and blocking the interactions with anti-csGRP78 autoantibodies etc).
Author Response
Q1: what about the specific/cross-reactivity of the described anti-csGRP78 autoantibodies? (I mean whether these autoantibodies cross-react with some other autoantigens different from GRP78?).
A1: The prostate cancer patient autoantibody has been extensively characterized (Ref. 23). The total anti-GRP78 IgG reacts almost exclusively with an epitope found in the N-terminal amino acid region Leu98-Leu115 (Ref. 23). We validated this specificity with anti-GRP78 IgG isolated from the serum of melanoma patients (Ref. 24). The GRP78 Ab isolated from RA patients reacts with an epitope in the C-terminal amino acid region Lys279-Ala298 (Ref, 54). The GRP78 Ab from T1D recognizes a citrullinated Arg510 in the C-terminal amino acid region Ile503-Glu514 (Ref, 83). The specificity of these autoantibodies appears to be specific for GRP78.
Q2: What happens with these autoantibodies after their binding to csGRP78? (whether the formed immune complexes are internalized into the target cell? If yes, what next?)
A2: There are limited reports addressing the fate of the csGRP78 immune complexes in the pathologies described in the manuscript. One report in the literature is that of the human monoclonal IgM SAM-6 antibody that once bound to csGRP78 is internalized and is finally responsible for a lethal accumulation of oxidized lipoproteins followed by apoptosis of the human pancreatic BxPC-3 cell line (Ref. 94). Extensive work in this field remains to be made.
Q3: csGRP78 is thought to play an important role in the pathogenesis-promoting activities of cancer cells and, particularly, cancer stem cells. How the described anti-csGRP78 autoantibodies affect the cancer-promoting function of csGRP78 in cancer cells and cancer-stem cells?
A3: CsGRP78 is a receptor with multiple functions where autoantibodies trigger multiple signaling pathways in the cells (Gonzalez-Gronow et al., IUMB Life 2021, 73:843-854). There is a large body of literature all referenced and discussed in the above review article from our laboratory. The basic mechanisms involve augmented cancer cell proliferation and cell-surface expression and secretion of GRP78, which acting in an autocrine fashion stimulate cell proliferation and autoimmunity. In cancer, or cancer stem cells, autoantibodies to the csGRP78 N-terminal region stimulate cell proliferation and viability; however, the autoantibodies to the GRP78 C-terminal region may induce cell apoptosis in autoimmune CNS diseases.
Q4: Use of exogenous csGRP78-neutralizing antibodies is discussed as a potential way for treating cancer (doi:10.3390/cells9040892; doi:10.1016/j.canlet.2021.10.004). Whether the presence of anti-csGRP78 autoantibodies will interfere with the use of exogenous (curative) antibodies to csGRP78?
A4: There are several exogenous antibodies with potential clinical interest against cancer (doi:10.1016/j.canlet.2021.10004). These antibodies have been tested in cancer cells and in xenograft mice models. All of them induce apoptosis and reduce cancer cell viability and none of them has been tested for their capacity to block the GRP78 autoantibodies described in this report. A recent study by Al-Hashimi et al. (doi:10.1200/JCO.2019.37.7-suppl.206) used a recombinant antibody (AEP858) in combination with enoxaparin as an antagonist of prostate cancer GRP78 N-terminal region autoantibodies, and observed a decrease in the induction of UPR and TF-procoagulation activity, abnormally elevated by the autoantibody. To date, the only GRP78 antibody evaluated in clinical trials is the IgM Ab SAM-6; however, the results did not show a significant clinical effect (doi:10.1097/CMR.ob013e328362cbc8). Although the limited reports are promising, an extensive work will be necessary before exogenous antibodies are safely used in clinical treatments.
Q5: If the authors consider the anti-csGRP78 autoantibodies as the pathogenic factors, they should discuss some potential ways to overcome such factors (e.g. exogenous antibodies or peptides binding csGRP78 and blocking the interactions with anti-csGRP78 autoantibodies, etc.)
A7: We added a new Section 7 to the manuscript where we discuss the evidence on exogenous Abs or peptides targeting csGRP78 as therapeutic agents. The low therapeutic significance of the Ab SAM-6 in clinical trials makes the subject very controversial. Until more evidence is produced, we prefer not to speculate on the use of these agents in cancer or autoimmune diseases.
Reviewer 2 Report
Overall, this review from Gonzalez-Gronow and Pizzo seeks to explore the roles of BiP autoantibodies in cancer and other diseases. It begins with a clear introduction that flows well into mechanistic insights of BiP autoimmunity. The diagram of the BiP IgG structure is simple but contains an appropriate amount of information.
My major concern is the section on BiP being a transmembrane protein. There is no credible evidence for this in the field. BiP does associate with membrane-associated proteins and is required for their folding but is definitely not a TM protein itself. Any reference to this and figure 3 need to be removed.
Overall, this review would be of only minor interest to the chaperone field or general readers.
Author Response
Q1: My major concern is the section of BIP being a transmembrane protein. There is no credible evidence for this in the field. BiP does associate with membrane-associated proteins and is required for their folding but is definitely not a TM protein itself. Any references to this and figure 3 need to be removed.
A!: CsGRP78 is not a transmembrane protein. We used a prediction model of BiP TM protein from an old paper (Redy et al., 2003, J. Biol. Chem., 278: 20915-20924) that was later corrected (Ref. 89). GRP78 is anchored to the cell surface via a GPI-protein link , associated to MTJ-3 and G∝q11 (Refs. 4, 89). Accordingly, any references to a possible GRP78 TM nature and Fig. 3 have been removed from the manuscript.
Reviewer 3 Report
The review “Physiological roles of antibodies to 78-kilodalton glucose-regulated protein (GRP78) in cancer and autoimmune disease” by Gonzalez-Gronow and Pizzo considers numerous issues related to the mechanisms of antibody formation under conditions of endoplasmic stress, anti-GRP78 antibody glycosylation, and the consequences of these processes for patients with cancer and various autoimmune diseases. The review seemed to me timely, interesting and deep, and, to be honest, I could not find any flaws with the exception of a couple of broken lines in the text (lines 163-164 and lines 298-299). To my opinion, the review deserves to be accepted for publication in Biomedicines.
Author Response
Q1: The review seemed to me timely, interesting and deep, and, to be honest, I could not find any flaws with the exception of a couple of broken lines in the text (lines 163-164 and lines 298-299)
A1: The broken lines have been fixed and minor changes in the text were made to answer the concerns of one of the reviewers. Also, Fig. 3 was removed in agreement with the recent information on the topography of csGRP78.
Reviewer 4 Report
In the present review by Gonzalez-Gronow and Pizzo entitled “Physiological roles of the autoantibodies to the 78-Kilodalton glucose-regulated protein (GRP78) in cancer and autoimmune diseases” the authors summarized the GRP78 role in several cancer and autoimmune diseases (e.g., MS, AMOGAD, NMO and LEMS).
Under ER stress GRP78 exits ER being translocated to plasma membrane where it stimulates the production of autoantibodies responsible for numerous pathologies.
Author Response
No concerns to be answered
Round 2
Reviewer 1 Report
Dear Authors,
Thanks for your responses and additions you made in the revised manuscript. I recommended to accept it for publication.
Good luck in your further research.
Author Response
Q1: no concerns
A1: no concerns to answer
Reviewer 3 Report
I had no serious objections at the first revision, and there are none now
Author Response
Q1: no serious concerns.
A1: no concerns to answer.